# The Impact of Altruistic Teaching on English as a Foreign Language (EFL) Learners’ Emotion Regulation: An Intervention Study

**DOI:** 10.3390/brainsci13030458

**Published:** 2023-03-08

**Authors:** Ali Derakhshan, Javad Zare

**Affiliations:** 1Department of English Language and Literature, Faculty of Humanities and Social Sciences, Golestan University, Gorgan 4913815759, Iran; 2Department of English Language and Literature, Faculty of Humanities, Kosar University of Bojnord, Bojnord 9453155168, Iran

**Keywords:** positive psychology, altruism, altruistic teaching, emotion regulation, EFL learners

## Abstract

The second language acquisition (SLA) field has recently seen heightened interest in the study and application of positive psychology (PP). Emotion regulation is one of the concepts that has been stressed in PP. Several studies in PP have delved into how controlling one’s emotions improves second language learning/teaching. One of the concepts that has slipped the minds of researchers in the field is altruistic teaching. Unlike egocentric acts, altruistic teaching acts are performed to improve others’ well-being. Despite their importance in causing positive emotional effects, no study has investigated the impact of altruistic teaching acts on learners’ emotion regulation. To bridge this gap, the present study sought to investigate the effect of learners’ altruistic teaching on their emotion regulation. The study followed a sequential explanatory comparison group pre-test–post-test design. One hundred forty-one English as a Foreign Language (EFL) learners were recruited for this intervention study and were divided into experimental and control groups. Learners in the experimental group performed altruistic teaching by teaching their peers how to write essays in English, whereas learners in the control group did group work tasks on English essay writing. The results of independent-sample *t*-tests and repeated-measures ANOVA showed that altruistic teaching significantly impacts EFL learners’ emotion regulation. The results of qualitative data pointed to five themes, including enjoyment, self-esteem, bonding, devotion, and progress. Overall, the results suggested that altruistic teaching impacts learners’ emotion regulation by enhancing their enjoyment, self-esteem, bonding, devotion, and progress. The paper has theoretical and pedagogical implications for SLA research and practice.

## 1. Introduction

Learning experiences are shown to be accompanied by a wide spectrum of pleasant and unpleasant feelings [1], and the language learning experience is by no means an exception. That is, language learners typically experience a variety of negative and positive emotions while acquiring a second/foreign language [2,3,4]. To function more effectively, learners need to manage the positive and negative emotions that come up at different phases of learning [5,6]. The mechanism of managing favorable and unfavorable emotions is technically called “Emotion Regulation” [7,8,9]. Emotion regulation generally pertains to “extrinsic and intrinsic processes that an individual goes through to evaluate, modify, or control his/her emotions to accomplish specific purposes and goals in life” ([10], p. 27). Taking this into the language learning domain, Bielak and Mystkowska-Wiertelak [11] described emotion regulation as psychological, behavioral, and cognitive processes language learners use to modulate their academic emotions. 

According to the school of “Positive Psychology” (PP), learners who can effectively sustain their positive feelings (e.g., enjoyment, happiness, etc.) and get rid of their negative feelings (e.g., boredom, anxiety, etc.) will outperform in academic contexts [12,13,14]. In light of PP, Piechurska-Kuciel [15] also mentioned that modulating academic feelings helps language learners attain the desired educational outcomes. Likewise, Li, Dewaele, and Jiang [16] noted that students who have control over their positive and negative emotions are more likely to succeed in educational settings. Notwithstanding the significant role of emotion regulation in language learners’ academic performance [13,15], little attention has been directed toward the predictors of this construct in language classes [17,18,19,20]. Moreover, PP variables influencing language learners’ emotion regulation have somehow been disregarded by previous scholars. To narrow this lacuna, this research endeavors to look into the potential impact of altruistic teaching, an important instance of PP constructs, on Iranian English learners’ emotion regulation. 

Altruistic teaching is theoretically underpinned by the ‘well-becoming through teaching/helping’ and the ‘teaching/helping rush’ hypotheses. The ‘well-becoming through teaching/helping’ hypothesis posits that learners become “well-becoming agents of change” if they are involved in wider social relationships ([21], p. 205). According to the theory, altruistic teaching, as it involves teaching others without egocentric motives, turns learners into agents of change by involving them in social relationships. More importantly, as it involves constant control and evaluation of one’s emotions, it is likely to impact learners’ emotion regulation. In this respect, Zare, Aqajani, and Derakhshan [22] explored the impact of altruistic teaching on learners’ emotions and summary writing skills, drawing on an experimental (comparison group, pre-test–post-test) and sequential explanatory design. The results of their analyses with 130 Iranian intermediate-level EFL learners pointed to a positive instructional influence on their emotions and summary writing skills by enhancing their self-esteem, gratitude, connectedness and community, happiness, and compassion. Additionally, according to the ‘teaching/helping rush’ hypotheses, tasks that draw individuals out of self-focus, such as altruistic teaching, help learners become fully engaged with material and others [23]. Such tasks help learners to become much more task-focused and less likely to experience negative emotions [23]. Hence, it is likely that altruistic teaching impacts learners’ emotion regulation. With these points in mind, the present paper sought to investigate the effect of learners’ altruistic teaching on their emotion regulation.

### 1.1. Altruistic Teaching

Altruism generally refers to activities that have no considerable benefit for the provider [24]. Pure altruism, as Gregersen [25] mentioned, is helping others without seeking any reward. According to Gregersen, MacIntyre, and Meza [26] “acting upon our altruistic tendencies enhances our self-esteem and makes us feel useful, offering a means to exercise our strengths and talents in meaningful ways” (p. 153). Considering the definition of altruism, Friedman [27] described altruistic teaching as instructing others without asking for anything in return. As put by Murphey [21], asking learners to teach their classmates gives them an appropriate opportunity to learn. He [23] further stated that putting learners in the teacher’s position helps learners master the course content in that it drives them to deeply engage with the learning materials. As pointed out by Buragohain and Senapati [28], altruistic teaching enables students to pay more attention to their classmates, form strong bonds with them, and develop a sense of belonging. This is why altruistic teaching directs learners toward pleasant emotions such as enjoyment, happiness, and fulfillment [22,28]. Despite the importance of altruistic teaching in educational environments and its salience in PP, a limited number of investigations have been conducted on this variable and its possible outcomes. The results of previous inquiries [22,29,30,31,32] uncovered that altruistic teaching has significant, favorable impacts on teachers, learners, and academic outcomes. 

### 1.2. Learner Emotion Regulation

There has been no agreement among academics and scholars on the description of emotion regulation. Thus, there are several statements on the definition of this concept. Cole, Michel, and Teti [33], for instance, defined emotion regulation as the ability to “respond to the ongoing demands of experience with the range of emotions in a manner that is socially tolerable and sufficiently flexible to permit spontaneous reaction as well as the ability to delay spontaneous reactions as needed” (p. 75). Further, Thompson [10] conceptualized this construct as the internal and external processes a person goes through to analyze, adjust, and manage his or her feelings. Concerning these descriptions, Asberg [34] characterized learner emotion regulation as the psycho-emotional processes learners go through to assess and navigate their academic emotions. As noted by Teng and Zhang [35], learners typically use a wide variety of strategies to up-regulate their pleasurable sensations, resulting in greater academic achievements [36]. They also employ several techniques to down-regulate their unpleasant feelings [37,38]. Through these techniques, they hinder the unfavorable effects of negative emotions on their classroom performance [39]. Thus, emotion regulation strategies help learners function more efficiently in the learning environment [14]. 

Due to the pivotal position of emotion regulation in students’ academic success [36,39], several researchers have empirically evaluated the effects of this variable in the classroom ecology (e.g., [40,41,42,43]). However, the determinants of this construct have been under-researched in both mainstream and language education [44,45,46]. Additionally, the possible impacts of PP variables, notably altruistic teaching, on learners’ emotion regulation have been overlooked in previous studies. To address the highlighted gaps, using a mixed-methods approach, this investigation sought to scrutinize the influence of altruistic teaching on Iranian English learners’ emotion regulation. The following question has guided the current investigation:

Does altruistic teaching affect Iranian English learners’ emotion regulation statistically significantly?

## 2. Methods

### 2.1. Setting and Design

The present analysis was conducted in an EFL context with EFL learners. A sequential explanatory comparison group pre-test–post-test design was followed to investigate the impact of altruistic teaching on learners’ emotion regulation. The comparison group pre-test–post-test design, which is a between-groups design, was followed to implement the intervention, the impact of altruistic teaching on learners’ emotion regulation. This between-group design was adopted to minimize the impact of demand characteristics on the results of the study. To this end, we did not inform the learners about the group they were assigned to. The design involved (1) pre-testing learners on their emotion regulation; (2) dividing them into experimental and control groups; (3) exposing the experimental group to the treatment, i.e., teaching peers altruistically how to write essays in English, and the control group to the placebo, i.e., doing group work tasks on English essay writing; and (4) post-testing them on their emotion regulation. Within the comparison group pre-test–post-test design, a sequential explanatory mixed-methods design was followed to investigate the impact of altruistic teaching on learners’ emotion regulation. Sequential explanatory designs require collecting and analyzing both quantitative and qualitative data, where qualitative data are used to expand on the results of quantitative data.

### 2.2. Participants

The participants of the study were 141 Iranian EFL learners. They were selected from 221 students in an English department with two undergraduate programs, i.e., B.A. in English Language Teaching and B.A. in English Language and Literature, at a state-run university. They all volunteered to participate in the study. First, their general English language proficiency level was measured through the Oxford Online Placement Test (OOPT). Next, learners with a B1 (intermediate) level of general English language proficiency, based on the Common European Framework of Reference for Languages (CEFR), were randomly selected through stratified sampling. This led to the selection of 141 learners whose L1 was Persian. They were studying either English language teaching (*n* = 57, 40.42%) or English language and literature (*n* = 84, 59.58%). Twenty-six learners were sophomores (18.44%); seventy-six learners were juniors (53.90%); and thirty-nine learners were seniors (27.66%). The participants included male (*n* = 49, 34.75%) and female (*n* = 92, 65.25%) learners with an age range of 20–24 (*M* = 22.48, *SD* = 1.26). They all participated in the study voluntarily and signed a written informed consent form, with the general purposes of the study highlighted in it.

### 2.3. Instruments

As part of the mixed-methods design, both quantitative and qualitative data were collected. Quantitative data involved using the OOPT and an emotion regulation questionnaire. Qualitative data included reflective frames and semi-structured interviews. Qualitative data were collected to increase the credibility and transferability of the analysis [47].

#### 2.3.1. OOPT

The OOPT is an online computer-adaptive placement test of English, designed for non-native speakers of English. It consists of two sections—Use of English and Listening. The Use of English section tests learners’ knowledge of grammar and vocabulary and the Listening section assesses their general listening ability. The test automatically marks each item and gives a score of 1–120. It reports learners’ English language proficiency at Pre-A1, A1, A2, B1, B2, C1, and C2 levels of the CEFR. It takes 45–60 min to complete.

#### 2.3.2. Emotion Regulation Questionnaire

The Emotion Regulation Questionnaire (ERQ) [8] was used to measure the learners’ emotion regulation. ERQ is a 10-item scale, developed to estimate learners’ inclination to regulate their emotions on a 7-point Likert scale, ranging from 1 (strongly disagree) to 7 (strongly agree). It measures learners’ emotion management in two aspects: (1) Cognitive Reappraisal and (2) Expressive Suppression. Cognitive Reappraisal deals with respondents’ emotional experience or what they feel inside. It includes six items. Expressive Suppression concerns respondents’ emotional expression or how they show their emotions in their talks, gestures, and behaviors. The internal consistency of the scale, i.e., Cronbach alpha, was computed as α = 0.84. The ERQ was designed through Google Forms and disseminated through Telegram.

#### 2.3.3. Reflective Frames

A reflective frame, which was designed through Google Forms and disseminated through Telegram, was used to collect data on the impact of altruistic teaching on learners’ emotion regulation. It included a Persian sentence prompt that asked learners to explain how altruistic teaching impacted their emotional experience and expression (*Teaching peers altruistically about how to write essays in English (didn’t) help(ed) me experience/express emotions …*). It was filled out by the learners once a week during the program.

#### 2.3.4. Semi-Structured Interview

Qualitative data also included semi-structured interviews with 18 learners (9 students from the experimental group and 9 students from the control group) in Persian, the respondents’ first language. These interviews were held one-on-one after the completion of the treatment. The purpose of the interviews was to find out how altruistic teaching impacted the respondents’ emotion regulation. A typical interview began with a statement of its purpose. Next, learners were familiarized with the concept of emotion regulation. Afterward, they were asked to answer the following questions: (1) *How did you like teaching classmates altruistically about essay writing in English?* (2) *Did you find it useful in managing your emotions?* and (3) *If so, in what ways did it affect your emotions and emotion regulation?* Finally, they were asked to share their concerns about the program. Each interview took 30 min on average.

### 2.4. Procedures

At first, learners who volunteered to participate in the study were asked to take the OOPT test. Based on their scores, learners with a B1 general English language proficiency level were recruited through a stratified random sampling procedure. Next, they were asked to fill out and sign a written informed consent form. The form outlined the purposes of the study very generally in order to ensure the study was not affected by demand characteristics. Those who volunteered to participate in the semi-structured interview were also asked to show their willingness by leaving their contact details. Next, as part of the comparison group pre-test–post-test design, learners were pre-tested on their emotion regulation with the ERQ [8]. The purpose of pre-testing participants was to make sure learners were uniform in terms of emotion regulation and English language proficiency before the treatment and that their difference would not affect their post-intervention emotion regulation. Afterward, they were randomly assigned to the experimental (73 students) and control (68 students) groups through stratified sampling. Their major, year of study, gender, and age were considered when assigned to the groups. To minimize the effects of demand characteristics, a single-blind design was adopted. Hence, the assignment of groups was concealed from the learners. The comparison group pre-test–post-test design, which is a between-groups design, was also adopted to ensure the study was not affected by demand characteristics. In this design, each participant receives only one independent variable treatment. Later, the experimental group underwent the treatment, and the control group underwent the placebo. Both the treatment and placebo involved a 10-session program (two sessions a week) of instruction on writing essays in English by one teacher, the second author of the paper. The overall structure of each session was the three “P”s approach, i.e., presentation, practice, and production. During the presentation phase, learners became familiar with the general structure of English argumentative essays and the steps for writing them. During the practice phase, learners were provided with annotated English argumentative essays to learn about the typical language forms of such essays. Last, during the production phase, they produced a sample of an English argumentative essay. Following each session, there was a practice session during which learners carried out supplementary activities on English essay writing in pairs. Besides doing these activities, the learners in the experimental group performed altruistic teaching during the practice sessions. Altruistic teaching involved teaching one’s group members altruistically how to write English argumentative essays. Hence, the points raised in each session of the English essay writing program were elaborated on by learners. Altruistic teaching was an outside-class experience, and the learners were free to do it online or in person based on their convenience. More importantly, they received no rewards whatsoever for doing it. They were also asked to report on the completion of altruistic teaching for each session. 

During the program, learners of both groups were asked to complete an online reflective frame every week. After the program, they were asked to complete the ERQ. The purpose of this administration of the ERQ was to investigate the effects of altruistic teaching on learners’ emotion regulation. After a one-week interval, Persian online semi-structured interviews were held with volunteers to shed light on how the program affected their emotion regulation.

### 2.5. Data Analysis

Analysis of the quantitative data, the ERQ, involved running independent sample *t*-tests, as the results of pre- and post-administration of the ERQ confirmed normality (*p* = 0.166 and 0.089 > 0.05), a prerequisite for using parametric tests. Moreover, repeated-measure ANOVAs were run for both the experimental and control groups to investigate if any statistically significant differences ensued from the pre- to post-administration of ERQ. All the analyses were carried out using the Statistical Package for Social Sciences (SPSS) software (Version 26). 

Analysis of the qualitative data, i.e., learners’ responses to the reflective frames and semi-structured interviews, was performed with MAXQDA (2022 version, VERBI, Berlin, Germany). After verbatim transcription, these data were checked for inconsistencies and incorrect spellings. Next, they were coded based on Braun and Clarke’s [48] six-step model of data analysis. This involved first familiarizing the coders with data. This step required repeated reading of the responses to grasp their breadth and depth. The next step was generating initial codes, which involved finding sections of data that seemed interesting. Searching for themes, the next step involved sorting codes into broader themes. Next, the derived themes were reviewed and refined. Later, the themes were defined and named. Finally, a report was prepared to describe the themes with anonymized data excerpts.

Regarding qualitative data analysis, different measures were taken to improve its trustworthiness [47]. These include considering credibility, transferability, dependability, and confirmability [49]. The credibility of the analysis was enhanced by data triangulation. The transferability of the analysis was considered by preparing a detailed report of the study’s setting, design, participants, instruments, and derived themes, accompanied by anonymized data excerpts. The dependability of the analysis was improved through member checking or participant validation [50]. To do so, we asked six respondents to see if the derived codes/themes reflected their opinions in the semi-structured interviews. The confirmability of the analysis [49] and researcher positioning were addressed by (a) recruiting two coders (both authors of the paper), (b) computing their inter-coder reliability (*α* = 0.85), and (c) asking a foreign colleague (from China) to code and check a portion of the data (*α* = 0.81). These measures helped us maintain both emic and etic perspectives throughout the analysis.

## 3. Results

### 3.1. Quantitative Data

Table 1 displays descriptive statistics for the pre- and post-administration of the ERQ. Here, CR represents the Cognitive Reappraisal facet; ES represents the Expressive Suppression facet; and ER represents overall emotion regulation. 

As Table 1 shows, the means for control group learners’ CR, ES, and ER were (*M* = 25.22, 13.14, 38.36, *SD* = 7.47, 4.59, 10.61, *N* = 68) on the pre-test and (*M* = 23.57, 13.30, 36.88, *SD* = 8.60, 4.99, 12.06, *N* = 68) on the post-test. On the other hand, the means for experimental group learners’ CR, ES, and ER were (*M* = 25.56, 13.61, 39.17, *SD* = 5.43, 3.83, 7.64, *N* = 73) on the pre-test and (*M* = 30.10, 18.21, 48.32, *SD* = 7.78, 5.94, 12.74, *N* = 73) on the post-test. As can be seen, the results point to an increase in the experimental group learners’ emotion regulation and a decrease in the control group learners’ emotion regulation. Independent-sample *t*-tests were run to investigate the statistical significance of these differences. Table 2 presents the results of these *t*-tests.

As Table 2 shows, pre-administration of the ERQ resulted in *t* values of 0.31, 0.66, and 0.52 and Sig. (2-tailed) values of 0.75, 0.51, and 0.60 for learners’ CR, ES, and ER, respectively. That is, there were no statistically significant differences between the control and experimental groups before the intervention (*p =* 0.75, 0.51, and 0.60 *>* 0.05). On the other hand, the post-administration of the ERQ pointed to the *t* values of 4.73, 5.29, and 5.46 and Sig. (2-tailed) values of 0.000 for learners’ CR, ES, and ER, respectively. In other words, there was a statistically significant difference between the control and experimental groups after the intervention (*p =* 0.000 *<* 0.05). In addition, the computing effect size for learners’ post-intervention CR, ES, and ER pointed to Cohen’s *d* values of 0.798, 0.892, and 0.921, which are all considered large effects, according to Plonsky and Oswald [51]. This means that the two groups were statistically significantly different in their CR, ES, and ER following the intervention. This is taken to mean that altruistic teaching improved learners’ emotion regulation significantly.

Repeated-measures ANOVAs were also run to investigate if any statistically significant differences ensued from the pre- to post-administration of ERQ. Table 3 and Table 4 present the results of repeated-measures ANOVAs for the control group and Table 5 and Table 6 display the results of repeated-measures ANOVAs for the experimental group.

As Table 3 shows, the repeated-measures ANOVA with a Greenhouse-Geisser correction did not show any statistically significant differences for control group learners (*F*(1.000, 67.000) = 0.563, *p* = 0.456 > 0.05).

**Table 4 brainsci-13-00458-t004:** Pairwise comparisons for the control group.

Measure: Emotion Regulation						
(I) Time	(J) Time	Mean Difference (I − J)	Std. Error	Sig.	95% Confidence Interval for Difference	
					Lower Bound	Upper Bound
1	2	75.007	1	75.007	0.563	0.456
2	1	8924.493	67	133.201		

As Table 4 shows, post hoc analysis with a Bonferroni adjustment did not point to a statistically significant difference in the control group learners’ emotion regulation from the pre-test to the post-test (*p* = 75.007 > 0.05). 

**Table 5 brainsci-13-00458-t005:** Tests of within-subjects effects for the experimental group.

Measure: Emotion Regulation							
Source		Type III Sum of Squares	df	Mean Square	*F*	Sig.	Partial Eta Squared
Time	Sphericity Assumed	3056.329	1	3056.329	28.539	0.000	0.284
	Greenhouse-Geisser	3056.329	1.000	3056.329	28.539	0.000	0.284
	Huynh-Feldt	3056.329	1.000	3056.329	28.539	0.000	0.284
	Lower-bound	3056.329	1.000	3056.329	28.539	0.000	0.284
Error (Time)	Sphericity Assumed	7710.671	72	107.093			
	Greenhouse-Geisser	7710.671	72.000	107.093			
	Huynh-Feldt	7710.671	72.000	107.093			
	Lower-bound	7710.671	72.000	107.093			

As Table 5 shows, the repeated-measures ANOVA with a Greenhouse-Geisser correction pointed to a statistically significant difference for experimental group learners (*F*(1.000, 72.000) = 28.539, *p* = 0.000 < 0.05).

**Table 6 brainsci-13-00458-t006:** Pairwise comparisons for the experimental group.

Measure: Emotion Regulation						
(I) Time	(J) Time	Mean Difference (I − J)	Std. Error	Sig.	95% Confidence Interval for Difference	
					Lower Bound	Upper Bound
1	2	−9.151	1.713	0.000	−12.565	−5.736
2	1	9.151	1.713	0.000	5.736	12.565

As can be seen in Table 6, post hoc analysis with a Bonferroni adjustment pointed to a statistically significant difference in the experimental group learners’ emotion regulation from the pre-test to the post-test (*p* = 0.000 < 0.05).

### 3.2. Qualitative Data

Analysis of the qualitative data, learners’ responses to the reflective frames and semi-structured interviews, led to the emergence of five themes, including (1) enjoyment; (2) self-esteem; (3) bonding; (4) devotion; (5) progress.

‘Enjoyment’ was the most frequently reported theme in learners’ responses to the reflective frames and semi-structured interviews (*n* = 34). As Gregersen, MacIntyre, and Meza [26] point out, altruistic acts make individuals happier. A strong sense of enjoyment is crucial for learning a second language [52]. Additionally, Shao, Pekrun, and Nicholson [53] found enjoyment positively correlated with motivation, self-regulation, and foreign language performance. According to Bryant and Veroff [54], enjoyment/happiness helps learners hold positive feelings and attitudes toward others and phenomena and thus enhances their capability to regulate negative emotions. In this respect, Tracey, a 22-year-old English learner, noted:(1)I didn’t like doing something for others. But when I realized how much progress I was making in English by teaching others, I tried to enjoy the experience and it worked. It really worked.

‘Self-esteem’ was the second most frequently mentioned theme in the learners’ responses to the reflective frames and semi-structured interviews (*n* = 29). Self-esteem is among the main components of the EMPATHICS model of PP [55]. As Gregersen, MacIntyre, and Meza ([26], p. 153) note, altruistic teaching “helps us value and feel appreciative of our own good fortune” and “enhances our self-esteem and makes us feel useful, offering a means to exercise our strengths and talents in meaningful ways”. Regarding self-esteem, Molly, a 20-year-old English learner, commented:(2)The program helped me control my anxiety. I am always nervous in English classes. At first, I didn’t see the point in doing it. But I had to do it so I decided to face my fear. Little by little, it increased my self-esteem because I was helping others learn. So, I faced my anxiety and controlled such emotions.

‘Bonding’ was another repeatedly mentioned theme by learners in their responses to reflective frames and semi-structured interviews (*n* = 26). Bonding/connectedness is one of the potential factors in PP [56,57]. One of the functions of positive emotions is that they lead to a social bond [58]. As altruistic behaviors are concerned with the well-being of other individuals, they promote a strong sense of connection with others and materials [23,26]. In this regard, Mia, a 23-year-old English learner, said:(3)It (the program) led to more connection and bonding between us, keeping us away from negative emotions.

‘Devotion’ to others was another frequently mentioned theme by learners (*n* = 21). Gregersen, MacIntyre, and Meza [26] see this as an emotional response shown by individuals when finding others in need of help. They [26] believe such empathic emotions bring altruistic motivation. In this respect, Bruce, a 21-year-old English learner, mentioned:(4)Watching others learn because of my help was amazing. It was like we were on the same ship and everyone had to do one’s part. When I saw that my friend needed help, I became more sensitive and devoted to the job.

Last but not least, in their responses to the reflective frames and semi-structured interviews, most learners mentioned ‘progress’ (*n* = 14). This is partly in keeping with the self-determination theory, which posits that self-determination or intrinsic motivation occurs only when one’s need for competence or a sense of perceived progress, connection with others, and autonomy or willingness to perform tasks is met. Altruistic acts seem to fulfill these three needs. In this sense, Catherine, a 20-year-old English learner, noted:(5)Because I am not very good at English, it was difficult for me to do the tasks, let alone help others. But there was a part of me that knew I could. So, I kept telling myself “I can do it. I will do my best”. I learned a lot and made much more progress than I could think of.

## 4. Discussion

The results resonate with the premises of PP in general; PERMA [57]; EMPHATICS [55]; the control-value theory [59]; the well-becoming through teaching/giving and teaching/helping rush hypotheses [23]. Generally, PP posits that positivity at three levels, including institutions, personality features, and emotions, leads to learners’ positive emotions and prosperity in different aspects of life, such as language learning [57,60]. Positivity in institutions means positivity at the group or social level [61], which corresponds to altruistic acts. In other words, consistent with the results of the present investigation, altruistic teaching or teaching without any egocentric motives results in emotion regulation and positive affective responses on the part of learners. Whereas positivity in personality features and emotions only leads to the individual’s well-being, positivity at the group level, i.e., altruistic teaching, improves both the individual’s and others’ well-being [26]. 

The results also pinpoint the principles of the PERMA model [57]. The PERMA model postulates that an interaction of positivity among the five elements of positive emotions, engagement, relationships, meaning, and accomplishment is necessary for an individual’s well-being to actualize [62]. That is, to prosper in different aspects of life, such as language learning, an individual (1) needs to invest emotionally, (2) be highly engaged in the activity, (3) be connected to others, (4) find meaning in life, (5) and achieve accomplishments. Altruistic teaching leads to people’s well-being by enhancing positive emotions such as enjoyment, compassion, empathy, devotion, self-esteem, gratitude, bonding, and connectedness, consistent with the results of this study and those of others [22,23,26], which necessitates exerting control over one’s emotions. Additionally, altruistic teaching leads to a high level of engagement with language learning tasks and people [23]. In this connection, Robinson and Tamir [63] compared task-focused and self-focused processing and concluded that task-focused processing is generally more likely to lead to positive emotions and mental health, and more desirable outcomes than self-focused processing. Murphey [23] extended this and assumed that involving learners in tasks that require helping/teaching others, i.e., altruistic acts, increases their engagement with materials and others. Regarding meaning and accomplishment in the PERMA model, altruistic teaching helps us find meaning in life by assisting others and value ourselves because of the feeling of accomplishment we experience when watching them to learn [26].

The results are also congruent with the EMPHATICS model. The EMPHATICS model is an extension of the PERMA model and considers the principles of complex dynamic systems, including situatedness, nestedness, interconnectedness, emergence, openness, self-modification, nonlinearity, dynamism, stability, multiple causes, multiplicity, and adaptiveness [64]. According to this model, there are 21 elements that contribute to one’s well-being. These include emotion, empathy, meaning, motivation, hope, optimism, resilience, agency, autonomy, time, hardiness, habits of mind, intelligence, identity, investment, imagination, character strengths, self-efficacy, self-concept, self-esteem, and self-regulation [64]. Among these elements, emotion, empathy, meaning, motivation, agency, autonomy, character strengths, self-efficacy, self-concept, self-esteem, and self-regulation are most relevant to altruistic teaching. Learners who undergo altruistic teaching are capable of managing their emotions; show empathic emotions towards others; find motivation and meaning in assisting others; are agents of their own actions and emotions; reach autonomy by teaching instead of just learning; find out about their character strengths by teaching others; are self-efficacious; develop self-concepts of their abilities; show high self-esteem; and are able to self-regulate their cognition, emotion, motivation, and learning [21,22,23,25,26].

The results are also compatible with the control-value theory (CVT). According to this theory, evaluations of control and value are determinants of achievement emotions [59]. Evaluation of control concerns self-perceived control over activities and evaluation of value deals with the value or meaning that one associates with achievement-related activities. The CVT posits that high perceived control and high positive value jointly cause pleasant achievement emotions, such as enjoyment, hope, and pride [59]. On the other hand, lack of control and high negative value jointly cause unpleasant achievement emotions, such as boredom, anxiety, and anger [59]. As the results suggest, altruistic teaching leads to positive achievement emotions [22] because learners who commit altruistic teaching have control over their actions and find meaning and value in helping/teaching and watching others learn. That is, they are capable of evaluating and managing their emotions. This underscores why altruistic teaching enhanced learners’ emotion regulation skills. 

The results also lend support to the well-becoming through teaching/giving hypothesis. This theory postulates not a steady state, but an “agentive action or activity that creates better well-being in others” for altruistic teaching ([21], p. 205). Hence, learners become “well-becoming agents of change” ([21], p. 205). That is, altruistic teaching involves constant control and evaluation of one’s emotions. This is why it enhanced learners’ emotion regulation significantly. Additionally, the theory posits that learners can learn more effectively and maintain their well-being in wider social relationships if engaged in altruistic teaching [23]. This is partly because altruistic teaching is like putting the learner in the driving seat. One learns better how to do something by doing it instead of watching it. 

The results also give credit to the teaching/helping rush hypothesis. The teaching/helping rush hypothesis posits that teaching/helping others learn causes learners to lose self-focus and enter “the territory of shared task-based joy and wonder” ([23], p. 339). This is why the results of the present analysis and prior research showed that altruistic teaching leads to happiness and enjoyment [22]. More importantly, tasks that draw individuals out of self-focus cause more success than self-focused ones [63]. In this respect, Robinson and Tamir ([63], p. 161) maintain that task-focused processing “is generally more conducive to positive affect, mental health, and desirable behavioral outcomes” than self-focused processing. Murphey ([23], p. 326) highlights that helping/teaching others helps learners “move to a deeper level of involvement” and “interact more with the material and others, which keeps them much more task-focused and less likely to slip into negative self-focusing”. That is, learners are less likely to experience negative emotions if they perform altruistic teaching. This underscores why altruistic teaching enhanced learners’ emotion regulation. In this connection, Zare, Aqajani, and Derakhshan [22] investigated the effect of altruistic teaching on learners’ emotions and their second-language summary writing skills. Their results pointed to the positive instructional influence of altruistic teaching on learners’ second-language summary writing skills by enhancing their self-esteem, gratitude, connectedness, happiness, and compassion.

The findings need to be treated with caution, considering the limitations of the study. First and foremost, collecting the data, both quantitative and qualitative, was mainly based on self-report measures, which are susceptible to limitations, such as honesty, social desirability, introspective ability, and response bias [65]. More importantly, the study was conducted over a short span. Longitudinal interventions are more likely to reveal the potential benefits of altruistic teaching. Finally, the study treated learners homogeneously. Considering learners’ individual differences in cognition and emotion is likely to lead to more fruitful findings. In this respect, Layous and Lyubomirsky [66] stress the importance of assessing the fit between the individual and the PP intervention in evaluating its effectiveness for each individual. Therefore, researchers are advised to consider these limitations in their future scholarly endeavors regarding altruistic teaching and emotion regulation.

## 5. Conclusions

Previous research shows that altruistic teaching leads to positive emotional effects on learners and enhances their L2 learning [23,26]. On the other hand, studies show that emotion regulation, or having control over positive and negative emotions, boosts success in educational contexts [16]. However, no empirical research, to the best of our knowledge, has investigated the association between altruistic teaching and emotion regulation. To this end, the present investigation explored whether altruistic teaching affects English learners’ emotion regulation statistically significantly, using a sequential explanatory comparison group pre-test–post-test design. The results of independent-sample *t*-tests pointed to a statistically significant difference between the control and experimental groups after the altruistic teaching intervention. Additionally, the results of repeated-measures ANOVAs showed a statistically significant difference in the experimental group learners’ emotion regulation from the pre-test to the post-test. On the other hand, the results of the thematic analysis of learners’ responses to reflective frames and semi-structured interviews pointed to five themes, including (1) enjoyment; (2) self-esteem; (3) bonding; (4) devotion; (5) progress. These results altogether mean that altruistic teaching significantly improved learners’ emotion regulation by enhancing their enjoyment, self-esteem, bonding, devotion, and progress in learning English.

The findings have both theoretical and pedagogical implications for SLA research and practice. Theoretically, the findings may be used as evidence for developing a theory of altruistic teaching and emotion regulation. To do so, the findings need to be used as a basis for comparative research across other variables and settings. Empirically, the findings may be used to inform materials development, teacher training, and language teaching in SLA. SLA material developers may develop tasks whose successful completion requires altruistic teaching. Moreover, the study informs SLA teacher trainers in training teachers to use altruistic teaching in their classes. Finally, SLA teachers may benefit from this study by learning how to effectively implement and manage altruistic teaching in their classes.

## Figures and Tables

**Table 1 brainsci-13-00458-t001:** Descriptive Statistics.

	Group	N	Mean	Std. Deviation	Std. Error Mean
CR (Pre)	Experimental	73	25.561	5.4313	0.6356
	Control	68	25.220	7.4750	0.9064
ES (Pre)	Experimental	73	13.616	3.8392	0.4493
	Control	68	13.147	4.5980	0.5576
ER (Pre)	Experimental	73	39.178	7.6491	0.8952
	Control	68	38.367	10.6124	1.2869
CR (Post)	Experimental	73	30.109	7.7881	0.9115
	Control	68	23.573	8.6080	1.0438
ES (Post)	Experimental	73	18.219	5.9400	0.6952
	Control	68	13.308	4.9903	0.6051
ER (Post)	Experimental	73	48.328	12.7497	1.4922
	Control	68	36.882	12.0639	1.4629

**Table 2 brainsci-13-00458-t002:** Independent-sample *t*-tests.

		Levene’s Test for Equality of Variances		*t*-Test for Equality of Means						
		F	Sig.	t	Df	Sig. (2-Tailed)	Mean Difference	Std. Error Difference	95% Confidence Interval of the Difference	
									Lower	Upper
CR (Pre)	Equal variances assumed	9.35	0.003	0.311	139	0.756	0.341	1.095	−1.823	2.506
	Equal variances not assumed			0.308	121.71	0.759	0.341	1.107	−1.850	2.532
ES (Pre)	Equal variances assumed	2.09	0.150	0.660	139	0.511	0.469	0.711	−0.937	1.876
	Equal variances not assumed			0.655	130.90	0.513	0.469	0.716	−0.947	1.886
ER (Pre)	Equal variances assumed	7.81	0.006	0.523	139	0.602	0.810	1.550	−2.254	3.875
	Equal variances not assumed			0.517	121.13	0.606	0.810	1.567	−2.293	3.914
CR (Post)	Equal variances assumed	0.50	0.477	4.733	139	0.000	6.536	1.380	3.805	9.266
	Equal variances not assumed			4.716	135.05	0.000	6.536	1.385	3.795	9.276
ES (Post)	Equal variances assumed	4.23	0.042	5.295	139	0.000	4.910	0.927	3.076	6.744
	Equal variances not assumed			5.327	137.57	0.000	4.910	0.921	3.087	6.732
ER (Post)	Equal variances assumed	0.37	0.539	5.467	139	0.000	11.446	2.093	7.306	15.58
	Equal variances not assumed			5.477	138.96	0.000	11.446	2.089	7.314	15.57

**Table 3 brainsci-13-00458-t003:** Tests of within-subjects effects for the control group.

Measure: Emotion Regulation							
Source		Type III Sum of Squares	df	Mean Square	*F*	Sig.	Partial Eta Squared
Time	Sphericity Assumed	75.007	1	75.007	0.563	0.456	0.008
	Greenhouse-Geisser	75.007	1.000	75.007	0.563	0.456	0.008
	Huynh-Feldt	75.007	1.000	75.007	0.563	0.456	0.008
	Lower-bound	75.007	1.000	75.007	0.563	0.456	0.008
Error (Time)	Sphericity Assumed	8924.493	67	133.201			
	Greenhouse-Geisser	8924.493	67.000	133.201			
	Huynh-Feldt	8924.493	67.000	133.201			
	Lower-bound	8924.493	67.000	133.201			

## Data Availability

The data presented in this study are available on request from the corresponding author. The data are not publicly available due to the ethical restrictions of the institution where the study took place.

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
