# Peer review of "The Impact of Altruistic Teaching on English as a Foreign Language (EFL) Learners’ Emotion Regulation: An Intervention Study"

_brainsci, 2023, doi:10.3390/brainsci13030458_

Round 1

Reviewer 1 Report

This is an interesting study on the effect of altruistic teaching on emotion regulation in EFL learners. I have several comments regarding the study:

1. First, there is a need for more literature review on altruistic teaching, how it works, its theoretical mechanisms as well as its relation to emotional regulation. The literature review seems limited

2. Second, in the method section, the authors need to clarify whether random assignment is used to assign the condition. Without random assignment, the study will be vulnerable to many potential confounds.

3. More demographic information regarding the participants will be useful.

4. How did the authors ensure that the study was not affected by demand characteristics?

5. There is very limited information regarding how the altrustic teaching intervention is administered. The method section must be detailed enough that it is sufficient for independent researcher to replicate the study

6. I would encourage the authors to share their data at OSF https://osf.io/ to ensure reproducibility

Author Response

Ms. Ref. No.:  brainsci-2230827
Title: The Impact of Altruistic Teaching on English as a Foreign Language (EFL) Learners’ Emotion Regulation: An Intervention Study
Brain Sciences

Dear Ms. Alice Wang,

Thank you again for considering our manuscript for publication in Brain Sciences. Our acknowledgment also goes to the anonymous reviewers for their nice words, valuable time, and informative feedback. Their feedback has considerably improved our paper. We have revised the paper accordingly. We hope our paper finds its place in your journal and attracts the attention of your readership.

Best regards,

Reviewer 1

Reviewer comment: This is an interesting study on the effect of altruistic teaching on emotion regulation in EFL learners. I have several comments regarding the study.

Response: Thank you.

Reviewer comment: 1. First, there is a need for more literature review on altruistic teaching, how it works, its theoretical mechanisms as well as its relation to emotional regulation. The literature review seems limited.

Response: Thank you. Unfortunately, the literature on altruistic teaching is very limited. There are only two studies that have been done on altruistic teaching by Murphey and Gregersen which are already cited. The theoretical background and the relationship between altruistic teaching and emotion regulation is added.

Reviewer comment: 2. Second, in the method section, the authors need to clarify whether random assignment is used to assign the condition. Without random assignment, the study will be vulnerable to many potential confounds.

Response: Done.

Reviewer comment: 3. More demographic information regarding the participants will be useful.

Response: Done.

Reviewer comment: 4. How did the authors ensure that the study was not affected by demand characteristics?

Response: Done. Please refer to the Procedures section. Three measures were taken to minimize the effects of demand characteristics: 1) deception, adopting a between-group design, and 3) selecting a single-blind design. First, the learners were informed about the purposes of the study only generally. Second, a comparison group pretest-posttest design which is a between-groups design was adopted where each participant received only one independent variable treatment. Finally, by adopting a single-blind design, the assignment of groups was concealed from the participants.

Reviewer comment: 5. There is very limited information regarding how the altrustic teaching intervention is administered. The method section must be detailed enough that it is sufficient for independent researcher to replicate the study.

Response: Done.

Reviewer comment: 6. I would encourage the authors to share their data at OSF https://osf.io/ to ensure reproducibility.

Response: Thank you.

Reviewer 2

Reviewer comment: Thanks for recommending me as a reviewer. In this paper sought to investigate the effect of learners’ altruistic teaching on their emotion regulation. The study followed a sequential explanatory comparison group pretestposttest design. English as a Foreign Language (EFL) learners were recruited for this intervention study and were divided into experimental and control groups. Learners in the experimental group performed altruistic teaching by teaching their peers how to write essays in English whereas learners in the control group did group work tasks on English essay writing. If the authors complete minor revisions, the quality of the study will be further improved.

Response: Thank you.

Reviewer comment: 1. The introduction section is well written. However, it is too short. If the authors describe the theoretical background related to altruistic teaching on english as a foreign language in more detail in the introduction section, it can help readers understand.

Response: Done.

Reviewer comment: 2. pg3: Authors should describe the characteristics of the subjects more specifically in the Methods section.

Response: Done.

Reviewer comment: 3. pg10: Authors should add about limitations in the Discussion section.

Response: Done.

Reviewer 2 Report

Thanks for recommending me as a reviewer. In this paper sought to investigate the effect of learners’ altruistic teaching on their emotion regulation. The study followed a sequential explanatory comparison group pretestposttest design. English as a Foreign Language (EFL) learners were recruited for this intervention study and were divided into experimental and control groups. Learners in the experimental group performed altruistic teaching by teaching their peers how to write essays in English whereas learners in the control group did group work tasks on English essay writing. If the authors complete minor revisions, the quality of the study will be further improved.

1. The introduction section is well written. However, it is too short. If the authors describe the theoretical background related to altruistic teaching on english as a foreign language in more detail in the introduction section, it can help readers understand.

2. pg3: Authors should describe the characteristics of the subjects more specifically in the Methods section.

3. pg10: Authors should add about limitations in the Discussion section.

Author Response

(The authors gave the same response as above.)

Round 2

Reviewer 1 Report

I appreciate the authors' revisoin. However, some of my previous comments were not properly addressed by the authors. For example:

1. There is still no elaboration on how altrustic teaching intervention was administered. This is problematic since altrustic teaching intervention is the main point of the study

2. The authors nicely mentioned about the random sampling procedure. However, my comments was on random assignment. The authors did not yet clarify whether random assignment was implemented.

3. There is also limited information regarding data availability.

Author Response

Ms. Ref. No.:  brainsci-2230827
Title: The Impact of Altruistic Teaching on English as a Foreign Language (EFL) Learners’ Emotion Regulation: An Intervention Study
Brain Sciences

Dear Ms. Simone Liu,

Thank you again for considering our manuscript for publication in Brain Sciences. Our acknowledgment also goes to the anonymous reviewer for his/her valuable time and feedback. We have done our best to address the reviewer’s concerns. Thank you.

Best regards,

Reviewer 1

Reviewer comment: I appreciate the authors' revision. However, some of my previous comments were not properly addressed by the authors. For example: 1. There is still no elaboration on how altruistic teaching intervention was administered. This is problematic since altruistic teaching intervention is the main point of the study

Response: Thank you. Done.

Reviewer comment: 2. The authors nicely mentioned about the random sampling procedure. However, my comments were on random assignment. The authors did not yet clarify whether random assignment was implemented.

Response: The participants were randomly assigned to the control and experimental groups through stratified sampling. Their major, year of study, gender, and age were considered when assigned to the groups, as highlighted in the Procedures section.

Reviewer comment: 3. There is also limited information regarding data availability.

Response: A data availability statement is added. As pointed out, the data presented in this study are available on request from the corresponding author. Yet, they cannot be made publicly available due to the ethical restrictions of the institution where the study took place.
